# Sample, time, and wording effects on estimating the prevalence of childfree adults: Insights from Japan

**Zachary P. Neal** [ID]*, **Jennifer Watling Neal** [ID]

Psychology Department, Michigan State University, East Lansing, MI, United States of America

* zpneal@msu.edu

**Data Availability Statement:** All data and materials necessary to replicate the results reported below are available at https://osf.io/wkeqp/.

## Abstract

Childfree adults neither have nor want children, but estimates of their prevalence vary widely, leading to ambiguity about how common this family status actually is. The goal of this study is to examine the effects of sample composition, time, and question wording on estimates of the prevalence of childfree adults. We pool 83 nationally representative estimates of the prevalence of childfree adults in Japan since 2000 using meta-regression to identify the influence of sex, marital status, year, and survey question. Prevalence estimates are higher when computed from samples of women than men, from samples of singles than married people, from samples collected more recently, and from surveys asking questions about expectations than wants. Most of the variation in estimates of the prevalence of childfree adults can be attributed to differences in sample composition, time, and question wording. Taking these factors into account, we estimate that over 2.5 million Japanese adults age 18-50, or 5.64% of this population, were childfree in 2020.

## Introduction

Childfree adults are adults who neither have nor want children [1]. Estimates of the prevalence of childfree adults vary widely, from lows between 2.2% and 9% [2–4] to highs between 17.6% and 27% [5, 6]. These variations are likely driven by differences in samples, times, and survey questions. However, representative demographic surveys that can identify childfree adults are rare, making it difficult to obtain a precise estimate or understand the role of these factors. Japanese demographic surveys that collect detailed information on fertility intentions and behaviors are plentiful as a result of concerns about the country's demographic trajectory, and they offer a unique opportunity to clarify these issues. In this study, we pool 83 estimates of the prevalence of childfree adults in Japan from five nationally representative surveys between 2000 and 2021, estimating the effect that sex, marital status, year of data collection, and survey question wording have on prevalence estimates.

**Funding:** This work was supported by awards to ZPN and JWN from the Japan Center and Asian Studies Center at Michigan State University. The funders had no role in study design, data collection and analysis, decision to publish, or preparation of the manuscript.

**Competing interests:** The authors have declared that no competing interests exist.

# Background

## Prevalence of childfree adults

Childfree adults have recently received attention in academic research [1, 7, 8] and the popular media [9–11]. Unlike parents, childfree adults do not have children, and do not want to have children. This attitude distinguishes childfree adults from non-parents who are planning to have children in the future (i.e., not-yet-parents), who are undecided about having children (i.e., undecided adults), or who wanted to have children but experienced medical or social barriers (i.e., childless adults) [1]. There is significant variation in the estimated prevalence of childfree adults, ranging between 2.2% and 27% in U.S.-based demographic studies [2–6].

This variation has led to ambiguity about how common childfree adults actually are. It is possible that this variation is simply due to random sampling error. However, it is also possible that this variation is driven by multiple features of a survey's design. First, the estimated prevalence of childfree adults may vary by *who* is included in the sample. For example, some studies restrict their universe to only women [2], only men [5], or only ever-married or partnered adults [4]. These restrictions in the gender or martial status of respondents may lead to variations in estimated prevalence because, for example, some studies have found that more men are childfree than women [6, 12], or that more singles are childfree than ever-married or partnered individuals [6, 12].

Second, the estimated prevalence of childfree adults may vary by *when* the data were collected. There are signs that the prevalence of childfree adults may be increasing over time. For example, in the United States, studies using panel data from the National Survey of Family Growth (NSFG) found that the percent of childfree men ages 15 to 49 increased from 9.9% to 20.2% between 2012 and 2018 [5] while the percent of childfree women ages 15 to 49 increased from 6% to 9.8% between 2010 and 2019 [13]. This suggests that more recent prevalence estimates may be larger than earlier prevalence estimates.

Third, the estimated prevalence of childfree adults may vary by *what* questions are asked. The WIDE typology to describe the different question wordings that can be used to identify childfree individuals [1]. First, *want* questions ask respondents whether they want or desire children. Second, *ideal* questions ask respondents to provide their ideal number of children. Third, *direct* questions ask respondents if they identify as childfree. Fourth, *expect* questions ask respondents whether they expect, intend, or plan to have children in the future. These different question wordings can have different implications for the prevalence estimates of childfree adults. For example, *expect* questions may not adequately distinguish between childfree adults and childless adults, yielding larger prevalence estimates than other question wordings.

## Insights from Japan

There is insufficient representative demographic data from the United States to formally test whether estimates of the prevalence of childfree adults vary by who, when, or what data were collected. However, demographic data from Japan provides a unique opportunity to examine the effect of these potential design characteristics. Driven by concerns about its rapidly declining fertility, both the Japanese government and independent researchers have conducted numerous demographic surveys that focus specifically on fertility behaviors and intentions. These surveys have separately summarized results by sex and marital status subgroups, have been repeated over several decades, and ask multiple questions about having children. Therefore, they have yielded a wealth of data that can be used to understand how estimates of the prevalence of childfree adults depend on who, when, and what data is collected.

We turn to Japanese demographic data to investigate these issues primarily because these data offer more detail about childfree adults than data from any other country. However, Japan is an interesting context in its own right for studying the childfree population. Japan has experienced a decline in fertility since the 1970s, [14–16]. This fertility decline has caused alarm about an aging population [17] and has inspired pronatalist policies and mass media framing designed to increase birth rates [16, 18, 19]. Against this pronatalist backdrop, being childfree remains highly stigmatized in Japan, particularly for women who report being pressured and questioned about their decision [16, 20, 21]. Nonetheless, few studies have explicitly attempted to estimate the prevalence of childfree adults in Japan or explore the role that they may play in these trends. One study used data from 2005 to examine non-parents' reasons for not having children, but concluded that it is "too early in Japan to argue that a child-free culture. . .is emerging" [14]. Therefore, while the primary aim of the current study is to understand how survey design characteristics are associated with the estimation of childfree prevalence, a secondary aim is to clarify the prevalence of childfree adults in Japan.

## Methods

We searched surveys conducted by Japanese government agencies and survey data archived with the Inter-university Consortium for Political and Social Research (ICPSR) to locate (a) nationally representative surveys that (b) included questions that can be used to identify childfree respondents. The first inclusion criterion excluded non-representative surveys because our goal is to obtain prevalence estimates that can be generalized to the population. The second inclusion criterion excluded surveys (e.g., East Asian Social Survey) that contained information on actual fertility, but not information on fertility intentions. These surveys could distinguish parents from non-parents, but were not able to distinguish childfree respondents from other types of non-parents. The second inclusion criterion also excluded surveys that asked about the ideal number of children for a hypothetical family, but not for the respondent themselves (e.g., Japan General Social Survey).

Within each of these surveys, we identified as childfree those respondents who (a) did not have children and (b) indicated that they did not want children, that they did not expect to have children, or that zero was the ideal number of children for them. From these identifications, we computed (when raw data was available) or extracted (when only cross-tabulations were available) the number of respondents identified as childfree, the number of respondents providing an answer to the relevant questions, and the number of respondents missing on the relevant questions. When sampling weights were available, we used the `survey` package for R [22] to estimate a population prevalence, then used the estimated population prevalence to compute a weight-adjusted number of childfree respondents in the sample. For each estimate, we also extracted the sex, marital status, and age range of the sample, the year of data collection, and the Japanese language question used to identify childfree respondents.

Following data extraction, we used the WIDE (want, ideal, direct, expect) framework to uniquely classify each survey question wording used to identify childfree respondents (see Table 1) [1]. First, we classified survey questions containing 欲, ほ, or 望 as 'want' type question wordings. Second, we classified survey questions containing 理想 as 'ideal' type question wordings. Third, we classified survey questions containing 予定, つもり, or なりそう as 'expect' type question wordings. None of the surveys asked 'direct' questions (e.g., "do you identify as childfree?"). When the original Japanese language instrument was not available ($N = 8$), question classifications were based on the English language documentation or codebook provided with the data.

**Table 1. Survey question wordings used to identify childfree respondents.**

| Type | Keyword | Japanese example | English translation | Survey[a] |
|------|---------|------------------|---------------------|-----------|
| **Want** | 欲 | 子どもが欲しいと思いますか | Do you want to have children? | LSA02 |
| | ほ | あなたは子どもがほしいですか | Do you want children? | NSFEC |
| | 望 | 希望する子どもの数 | Desired number of children | JNFS |
| **Ideal** | 理想 | あなた方ご夫婦にとって理想的な子どもの数は何人ですか | What is the ideal number of children for you and your spouse? | JNFS |
| **Direct** | – | Not asked | – | – |
| **Expect** | 予定 | 今後お子さんをお持ちになる予定はありますか | Do you plan to have children in the future? | NHS |
| | つもり | あなた方ご夫婦はこれから何人子どもを生むつもりですか | How many children do you and your spouse plan to have? | JNFS |
| | なりそう | 実際になりそうなあなたの人生はどのタイプですか (結婚するが子どもは持たず) | What type of life do you expect to have? (get married but have no children) | JNFS |

[a] LSA02 = Longitudinal Survey of Adults in the 21st Century, 2002 cohort; NSFEC = National Survey on Family and Economic Conditions; JNFS = Japanese National Fertility Survey; NHS = National Household Survey.

To examine the effect of sex, marital status, age range, year of data collection, and survey question wording on the prevalence of childfree adults, we estimated a random-effects meta-regression via a generalized linear mixed model [23, 24] using the `metafor` package for R [25]. Because the effect size of interest is a proportion, we first applied a logit transformation. We use a multi-level specification, in which estimates are nested in surveys, to account for possible shared instrument variance [26]. Additionally, we specify an estimated variance-covariance matrix that accounts for possible autocorrelation in estimates obtained from the same sample of individuals in longitudinal panels [27]. Men, married respondents, and want-type question wordings serve as the reference categories for categorical variables measuring sex, marital status, and survey question wording, respectively. The year variable is centered on 2000.

Some estimates were computed from samples with a wide age range (e.g., JNFS in 2005, ages 18-50), while others were computed from samples with narrower age ranges that are younger (e.g., LSA12 in 2013, ages 21-30) or older (e.g., LSA02 in 2011, ages 30-44). To account for the variation in age ranges across samples, we control for each sample's minimum and maximum age, centering these variables on 18 and 50, respectively.

To evaluate the sensitivity of the meta-regression to these model specifications, we also estimated a corresponding 'naive' model using OLS regression to predict extracted prevalence values as a function of the same independent variables, but without considering possible shared instrument variance or panel autocorrelation. This naive model yielded similar coefficients with an identical pattern of statistical significance, indicating that the findings reported below are not a product of model specification.

The Michigan State University Institutional Review Board determined this study to be 'not human subjects' on 25 April 2023 (STUDY00009134) because the data are existing, public, and de-identified. All data and materials necessary to replicate the results reported below are available at https://osf.io/wkeqp/.

## Results

Fig 1 summarizes our data search and extraction process. We initially identified 20 nationally representative surveys from government ($N = 15$) and ICPSR ($N = 5$) sources. After reviewing

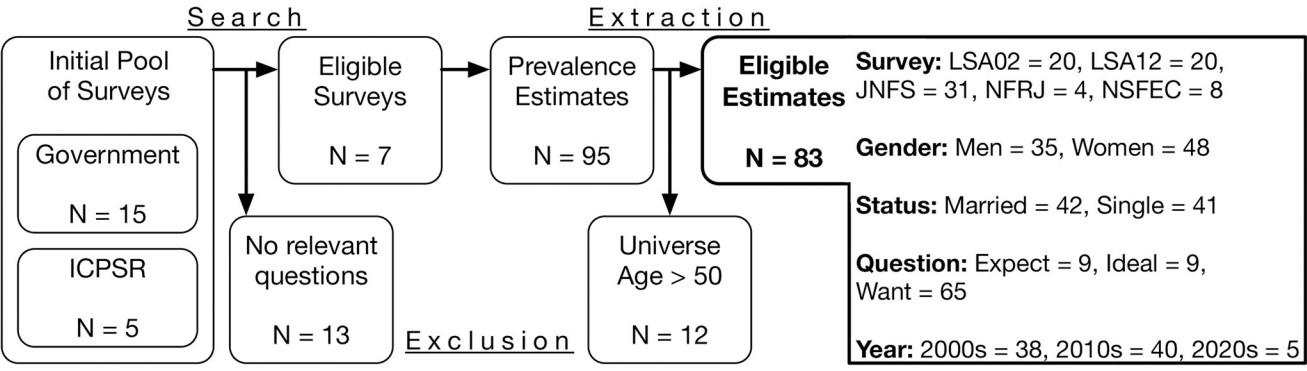

**Fig 1. Summary of data search and extraction.**

the surveys' codebooks, 13 were excluded because they did not contain questions that could be used to identify childfree respondents. From the remaining 7 eligible datasets, we extracted 95 estimates of the prevalence of childfree adults in Japan. To ensure that these estimates were computed for reasonably similar age universes, we excluded 12 estimates computed from samples containing respondents older than 50, yielding a final pool of 83 prevalence estimates that summarize 253,900 observations. Similar to the wide variation observed in US-based estimates, these estimates range from 1.24% to 40.31% ($M = 8.21$, $SD = 7.45$).

These estimates come from five nationally representative surveys: Japanese National Fertility Survey (JNFS), Longitudinal Survey of Adults in the 21st Century 2002 cohort (LSA02), Longitudinal Survey of Adults in the 21st Century 2012 cohort (LSA12), National Family Research of Japan (NFRJ), and National Survey on Family and Economic Conditions (NSFEC). Details of each survey's sampling methodology is provided in the *Supplementary Materials* at https://osf.io/wkeqp/. Each estimate captures the prevalence of childfree adults in a sample composed of respondents with a given sex and marital status, estimated by asking a given question in a given year. Two-thirds of the samples used to compute these estimates (55 of 83) had less than 5% missingness. On average these samples were missing data on 4.02% of respondents ($SD = 4.02$, Range = 0% and 25.71%). A table summarizing every included prevalence estimate is available in the *Supplementary Information* at https://osf.io/wkeqp/.

Table 2 reports the results of a multi-level random-effects meta-regression predicting estimates of childfree prevalence in Japan as a function of sex, marital status, time, and survey question wording, controlling for the minimum and maximum age in the sample. This

**Table 2. Meta-regression predicting estimates of childfree prevalence in Japan.**

|  | B | SE | p | OR |
|---|---|---|---|---|
| Intercept | -4.403 | 0.205 | 0 | 0.012 |
| Women | 0.18 | 0.080 | 0.025 | 1.198 |
| Single | 1.549 | 0.092 | < 0.001 | 4.704 |
| Year | 0.027 | 0.007 | < 0.001 | 1.027 |
| Expect question | 0.925 | 0.138 | < 0.001 | 2.521 |
| Ideal question | 0.118 | 0.139 | 0.397 | 1.125 |
| Minimum Age in Sample | 0.053 | 0.017 | 0.003 | 1.054 |
| Maximum Age in Sample | -0.017 | 0.009 | 0.063 | 0.983 |
| $R^2$ | 0.902 | | | |

model's AIC (54.65) and BIC (77.82) are much lower than in a reduced model with no independent variables (211.64 and 218.86, respectively), indicating that the model offers an excellent fit to the data. The $R^2$, computed as the squared correlation between the observed and predicted prevalence estimates, indicates that these variables account for over 90% of the variation in the extracted prevalence estimates. Because the estimated coefficients (B) are reported as log odds, for interpretability we also report the odds ratio (OR). The intercept indicates that the estimated prevalence of married men in a sample of 18-50 year old respondents in 2000 is 1.2%.

With respect to sex, we find that samples of women yield childfree prevalence estimates that are about 20% ($OR = 1.198$, $p = 0.025$) higher than samples of men. This differs from US-based estimates, which has found that men are more likely to be childfree [6, 12].

With respect to marital status, we find that samples of singles yield childfree prevalence estimates that are nearly 5 times ($OR = 4.704$, $p < 0.001$) higher than samples of married people. This is consistent with US-based estimates, which also suggest singles are more likely to be childfree [6, 12].

With respect to time, we find that samples collected more recently yield higher childfree prevalence estimates. Specifically, each year beyond 2000 is associated with a 2.7% increase in the estimated prevalence of childfree adults ($OR = 1.027$, $p < 0.001$). This is consistent with trends of the increasing prevalence of childfree women [13] and men [5] in the United States.

With respect to question wording, we find that surveys asking expect questions yield prevalence estimates that are more than twice as large as surveys asking want questions ($OR = 2.521$, $p < 0.001$). In contrast, we find that surveys asking ideal questions yield prevalence estimates that are not statistically significantly different from surveys asking want questions ($OR = 1.125$, $p = 0.397$). This is consistent with expectations that 'expect' type questions could yield overestimates by conflating childfree and childless respondents, and that 'want' or 'ideal' type questions yield more accurate estimates of prevalence [1].

To contextualize these results, we can combine the model estimates with population counts from the 2020 Japanese census to estimate the number and prevalence of childfree Japanese adults age 18-50 in 2020. These estimates suggest that 9.1% ($N = 1,011,582$) of single Japanese men, 2.08% ($N = 241,507$) of married Japanese men, 10.71% ($N = 962,240$) of single Japanese women, and 2.48% ($N = 338,732$) of married Japanese women were childfree in 2020. Collectively, this suggests that over 2.5 million Japanese adults age 18-50, or 5.64% of this population, were childfree in 2020.

## Conclusion

Childfree adults neither have nor want children [1]. Although this group is believed to be large and growing, attempts to estimate its size have yielded estimates varying from 2.2% and 27% of the population [2–6]. In this paper, we use the wealth of representative data on Japanese adults' fertility intentions and behaviors to examine how the estimated prevalence of childfree adults depends on a survey's design characteristics, including *who* was included in the sample, *when* the data was collected, and *what* questions were asked. We use a multi-level meta-regression to examine 83 unique prevalence estimates extracted from five nationally-representative surveys of over 250,000 adults age 18-50. We find that estimates of childfree prevalence are higher in samples of women than men, in samples among singles than married people, in samples collected more recently, and in samples asked questions about expectations than wants or ideals. Taking into account these sources of variation in estimates of the prevalence of childfree adults, these findings suggest that over 2.5 million Japanese adults age 18-50, or 5.64% of this population, were childfree in 2020.

These results shed light on why prior estimates of the prevalence of childfree adults exhibit so much variation, and have implications for designing research on the childfree population. Like prevalence estimates in the US, estimates in Japan vary widely, but we find that the majority of this variation can be explained by who is included in the sample, when the sample was collected, and what questions were asked. Therefore, prevalence estimates should be interpreted narrowly as describing only the sex(es) and marital status(es) from which the data were collected, and only at the time they were collected. Additionally, to avoid overestimation, prevalence estimates should rely on questions about whether the respondent wants, not expects to have, children.

More narrowly, these results also have implications for Japanese demographic policy. Driven by a long-term decline in fertility rates [14–16], government policy and rhetoric is often focused on encouraging adults to have children [16, 18, 19]. These efforts may have an impact on adults who are undecided about whether they want children. However, they are unlikely to be effective for the large number of adults who simply do not want children. Moreover, in cases where pronatal policies do encourage adults who do not want children to have children anyway, such policies risk trading a demographic crisis for a physical and mental health crises [28, 29].

These results must be viewed in light of some limitations, which identify directions for future research. First, these results are based on data from a single country. We focus on Japan because it offers more detailed representative data about fertility intentions than other countries. However, future studies should explore these design effects on prevalence estimates in other countries, or explore the feasibility of a cross-national meta-analysis that explicitly models country-level effects. Second, because most data are available only as cross-tabulations, these models consider only a limited set of demographic characteristics (sex, marital status). Future studies should explore the potential effect of other demographic characteristics such as age or education, which may be possible to the extent that raw micro-data rather than cross-tabulated data is available. Third, although definitions of 'childfree' vary in whether they are restricted to biological children [2] or both biological and non-biological (e.g., adopted, foster, and step-) children [1], we are unable to test whether such differences in definition affect prevalence estimates because the questions on these surveys do not explicitly indicate whether 'children' includes only biological children, or both biological and non-biological. It is rare for adults to have only non-biological children, so the scope of the definition likely has limited effect on the prevalence estimates of childfree adults. However, future studies should explore the potential effect. Finally, although we find that the prevalence of childfree adults has increased over the past 20 years, we cannot determine whether these temporal changes are driven by age, period, or cohort effects. Again, the availability of micro-data, especially on respondents' age and birth year, may help clarify this.

These limitations notwithstanding, this is the most comprehensive attempt to estimate the prevalence of childfree adults in a given country. By pooling many independent prevalence estimates collected from a range of samples, at a range of times, using a range of methods, we can both identify the effect of these design characteristics on the estimates, and obtain a more precise estimate of prevalence than is possible from any single survey. In Japan, an increasing number of both single and married Japanese men and women do not want children, with single women leading the trend; over 10% of single Japanese women age 18-50 did not want children in 2020. More broadly, research focused on studying the childfree population in Japan and elsewhere must attend to the effect that sample composition, time, and question wording can have on estimates of prevalence.

## Author Contributions

**Conceptualization:** Zachary P. Neal, Jennifer Watling Neal.

**Data curation:** Zachary P. Neal, Jennifer Watling Neal.

**Formal analysis:** Zachary P. Neal, Jennifer Watling Neal.

**Funding acquisition:** Zachary P. Neal, Jennifer Watling Neal.

**Investigation:** Zachary P. Neal, Jennifer Watling Neal.

**Methodology:** Zachary P. Neal, Jennifer Watling Neal.

**Writing – original draft:** Zachary P. Neal, Jennifer Watling Neal.

**Writing – review & editing:** Zachary P. Neal, Jennifer Watling Neal.

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
