## [Decision Letter · Decision Letter 0]

26 Jan 2024

PONE-D-23-35738Sample, time, and wording effects on estimating the prevalence of childfree adults: Insights from JapanPLOS ONE

Dear Dr. Neal,

Thank you for submitting your manuscript to PLOS ONE. After careful consideration, we feel that it has merit but does not fully meet PLOS ONE’s publication criteria as it currently stands. Therefore, we invite you to submit a revised version of the manuscript that addresses the points raised during the review process.

The manuscript has been evaluated by one reviewer and their comments are available below. The reviewer has raised a number of concerns. They request improvements to the reporting of methodological aspects of the study, for example, regarding the age ranges of the populations and sampling methods of the surveys. The reviewer also notes that the supplementary material mentioned in the manuscript was not present, please ensure this is uploaded in the revised version.

Could you please carefully revise the manuscript to address all comments raised?

Please note that we have only been able to secure a single reviewer to assess your manuscript. We are issuing a decision on your manuscript at this point to prevent further delays in the evaluation of your manuscript. Please be aware that the editor who handles your revised manuscript might find it necessary to invite additional reviewers to assess this work once the revised manuscript is submitted. However, we will aim to proceed on the basis of this single review if possible. 

We look forward to receiving your revised manuscript.

Kind regards,

Laura Kelly

Division Editor

PLOS ONE

Journal Requirements:

2. Thank you for stating the following financial disclosure: "This work was supported by awards to ZPN and JWN from the Japan Center and Asian Studies Center at Michigan State University. The funders played no role in the study."

Reviewers' comments:

Reviewer's Responses to Questions

**Comments to the Author**

1. Is the manuscript technically sound, and do the data support the conclusions?

Reviewer #1: Yes

2. Has the statistical analysis been performed appropriately and rigorously? 

Reviewer #1: Yes

3. Have the authors made all data underlying the findings in their manuscript fully available?

Reviewer #1: No

4. Is the manuscript presented in an intelligible fashion and written in standard English?

Reviewer #1: Yes

5. Review Comments to the Author

Reviewer #1: Thanks for the opportunity to review this well-written paper. As fertility rates are decreasing in most high-income countries, more individuals choose not to have children, known as “childfree”. However, due to the difficulty of measuring “intention”, rare nationally representative surveys exist to estimate the prevalence of childfree adults, and the limited existing numbers are highly varied due to differences in sample characteristics, survey time, and other reasons. The authors chose Japan (which has rich relevant survey data) as an example to understand some key factors influencing the prevalence of childfree adults and gave an overall estimate. This paper set the scene for future studies on how to estimate and report the prevalence of childfree adults. However, some aspects should be considered before accepting this paper.

1. The definition of “childfree adults”.

Although the authors gave a clear definition of “childfree adults” (i.e., adults who neither have nor want biological, adopted, foster, or step-children), some cited literature (for example, citation 2: Abma JC, Martinez GM. Childlessness among older women in the United States: Trends and profiles. Journal of Marriage and Family. 2006;68(4):1045–1056) only considered biological children when they defined different types of childlessness, which is inconsistent with the authors’ definition. I suggest the authors to double check how “childfree” was defined in their citations, and only cite those using the same definition or clarify the differences in the definition.

Also, for the question wordings presented in Table 1, it may not be appropriate to assume that “children” equals to “both biological and non-biological children”. Although not explicitly stated, the questions like “do you want to have children” or “do you plan to have children in the future” may imply a focus on biological children, as the authors commented in another paper (https://doi.org/10.31234/osf.io/fa89m). The authors should discuss these nuances unless the Japanese surveys explicitly define children as both biological and non-biological.

2. It would be helpful if the authors could provide the demographic characteristics and sampling methods of the five national surveys and raw data of the 83 estimates.

3. Do all the 83 estimates come from populations aged 18 to 50, or are some from younger (e.g., 18-30) and some from older (e.g., 30-50) age groups? Different age ranges may confound the results.

4. The authors mentioned “A forest plot summarizing every included prevalence estimate is

available in the supplementary materials”, but I didn’t find any supplementary materials.

6. PLOS authors have the option to publish the peer review history of their article (what does this mean?). If published, this will include your full peer review and any attached files.

Reviewer #1: **Yes: **Chuyao Jin

---

## [Decision Letter · Decision Letter 1]

28 Mar 2024

Sample, time, and wording effects on estimating the prevalence of childfree adults: Insights from Japan

PONE-D-23-35738R1

Dear Dr. Neal,

We’re pleased to inform you that your manuscript has been judged scientifically suitable for publication and will be formally accepted for publication once it meets all outstanding technical requirements.

Kind regards,

Ranjan Kumar Prusty, Ph.D.

Academic Editor

PLOS ONE

Additional Editor Comments (optional):

"I agree with the reviewer that it’s a well-written paper with an emerging idea. However, I have some queries and comments on this paper that the authors may look into:

The study defines childfree adults as 'adults who neither have nor want children.' However, they are potential childfree adults. They can bear a child in the future unintentionally or with a change in intention. I would like to know the authors' view on this.

Is there any spatial difference in such a concept? Please clarify if there are any geographical differences in the selected study.

I was interested in knowing the reasons for such intentions of couples. Of course, I see you have mentioned it as an emerging concept, but a micro-level study would be interesting."

Reviewers' comments:

Reviewer's Responses to Questions

**Comments to the Author**

1. If the authors have adequately addressed your comments raised in a previous round of review and you feel that this manuscript is now acceptable for publication, you may indicate that here to bypass the “Comments to the Author” section, enter your conflict of interest statement in the “Confidential to Editor” section, and submit your "Accept" recommendation.

Reviewer #1: (No Response)

2. Is the manuscript technically sound, and do the data support the conclusions?

Reviewer #1: Yes

3. Has the statistical analysis been performed appropriately and rigorously? 

Reviewer #1: No

4. Have the authors made all data underlying the findings in their manuscript fully available?

Reviewer #1: Yes

5. Is the manuscript presented in an intelligible fashion and written in standard English?

Reviewer #1: Yes

6. Review Comments to the Author

Reviewer #1: (No Response)

7. PLOS authors have the option to publish the peer review history of their article (what does this mean?). If published, this will include your full peer review and any attached files.

Reviewer #1: **Yes: **Chuyao Jin

---

## [Editor Report · Acceptance letter]

2 Apr 2024

PONE-D-23-35738R1 

PLOS ONE

Dear Dr. Neal, 

I'm pleased to inform you that your manuscript has been deemed suitable for publication in PLOS ONE. Congratulations! Your manuscript is now being handed over to our production team.

Kind regards, 

on behalf of

Dr. Ranjan Kumar Prusty 

Academic Editor

PLOS ONE